# Problems of Multiscale Brittleness Estimation for Hydrocarbon Reservoir Exploration and Development

**Nikita Dubinya [1,2,*], Irina Bayuk [1,*] and Milana Bakhmach [1]**

[1] Laboratory of Fundamental Problems of Petroleum Geophysics and Geophysical Monitoring, Schmidt Institute of Physics of the Earth of the Russian Academy of Sciences, 123242 Moscow, Russia; mbakhmach@ifz.ru

[2] Science and Technology Center of Geophysics and Mineral Resources, Moscow Institute of Physics and Technology, National Research University, 141701 Dolgoprudny, Russia

[*] Correspondence: dubinia.nv@mipt.ru (N.D.); ibayuk@ifz.ru (I.B.)

**Abstract:** The study is focused on the problem of using geophysical data to estimate brittleness of rock masses for the needs of petroleum industry. Three main developed ways to estimate brittleness—mineral-based, log-based, and elastic-based brittleness indices—are discussed from the perspective of scaling factor. The study highlights the contradictions between brittleness indices calculated from the same data using various ways of introducing brittleness. These contradictions are explained by scaling factor, as geophysical data used for brittleness estimation are typically obtained at different spatial and temporal scales. A model based on the effective medium theory is used to understand the relationships between inner structure of inhomogeneous rocks and their brittleness indices estimated from laboratory tests on core samples as well as log data analysis.

**Keywords:** brittleness; brittleness index; reservoir geomechanics; hydraulic fracturing; unconventional reservoirs; effective medium theory; rock physics modeling

## 1. Introduction

Exploration and development of hydrocarbon reservoirs are related to many fields of applied sciences and mechanics in particular. Filtration processes, well drilling, well log data obtaining and assessment, and hydraulic fracturing are all closely related to hydrodynamics and rock mechanics [1]. It is natural that methods used in these fields are being developed to answer industrial needs: for example, failure mechanics has been developed a lot to understand and optimize hydraulic fracturing procedures. Consideration of hydrocarbon fields' exploration and development makes emergence of specific problems evident. Some of these problems have their analogs in the other fields, so ideas regarding the ways to solve them are available. Nevertheless, some problems not typical for other branches of science arise as well, leading to the need of creating new principal approaches to deal with them. The current study is aimed at one of such problem—rock brittleness estimation.

Brittleness of the medium is a very complicated topic in its nature [2–5]. Many mechanical concepts have clear physical and mathematical meaning: elastic moduli are defined with respect to stress and strain changes; strength properties are defined as conditions standing for failure, and so on. Brittleness is somewhat unique among these concepts. It is generally easy to understand what brittleness is in general. Comparison of two materials, for instance, glass and clay, leads to a clear understanding that one of them (glass) is brittle, and the other one (clay) is not [6]. Brittleness has been historically introduced in many different ways, including qualitative definitions, such as lack of ductility [6,7], rock failure with lack of preceding plastic flow [8], and destruction of internal cohesion [9]. Nevertheless, whenever one tries to introduce brittleness as a certain physical property with mathematical definition, a number of problems become evident [2]. While the example of glass and clay reveals a clear distinction between brittle and ductile material, it is very

difficult to claim that glass is several times more brittle than clay and answer how many times in particular. Lack of a strict mathematical definition of brittleness brings brittleness discussions from the quantitative to the qualitative plane [9]. While qualitative discussions of brittleness and drawn conclusions may be enough for some practical problems, the other problems highlight the need to develop mathematical approaches to evaluate brittleness. Hydraulic fracturing is among such problems: it is preferable to initiate hydraulic fracture growth in a brittle zone of a hydrocarbon reservoir, but not in a ductile zone [5]. Given the complicated conditions of hydrocarbon reservoirs being in development nowadays, brittle zones should be analyzed with numbers, but not with qualitative discussions [5]. As a result, numerous studies are carried out to develop mathematical ways to estimate brittleness of rock masses from data obtained from hydrocarbon fields [3–5].

Quantitative definitions of rock brittleness have been developed a lot since the first concepts mentioned above, as new techniques of dealing with stress–strain deformation curves characterizing various materials have been emerging. Such concepts as total strain at the point of failure [2,10] and ratio between reversible and total energy at failure [11,12] were proposed to overcome the problem of quantitative brittleness evaluation. Although these definitions are mathematically correct and have solid physical background, a problem arises from of the need to analyze deformation curves. Keeping in mind that hydrocarbon reservoirs are the focus of many practical studies, deformation curves can only be obtained from laboratory tests on extracted rock samples. At the same time, rock sample experiments are complicated and costly, so alternative ways of dealing with brittleness have been developed with regard to various geophysical data.

Field data on rock masses may be generally divided into three categories with respect to their source. Data on rock mass physical properties may be obtained from seismic surveys, well logs, and laboratory experiments on core samples. These three sources represent three distinct spatial scales: seismic surveys make it possible to deal with rocks physical properties averaged for volumes compared to seismic wavelength (from 10 to 100 m); well logs characterize the lower scales of the first meters; finally, special experiments on rock samples can be carried out at the lowest scale—from microns to centimeters, depending on the kind of experiment. As far as rock masses are heterogeneous, the same properties are different at each of these scales [13]. There are special techniques, covered by the general approach known as 'rock physics modeling' aimed at construction of mathematical models describing the scaling of effective properties of heterogeneous rock masses with respect to this effect [14]. Once again, these methods are developed to deal with widely used mechanical properties, such as elastic moduli, but not with brittleness.

Heterogeneity of rocks results in the inability of the best techniques to estimate brittleness from deformation curves to characterize brittleness at any scale aside from the lowest one. The main idea of alternative methods of brittleness evaluation from geophysical data was to avoid laboratory tests and estimate brittleness from well logs. Mews et al., 2019 [4] proposed a general classification of developed methods of estimating brittleness with respect to the data used for calculation. Three groups of brittleness indices (BI) were introduced: mineral-based brittleness indices (MBI), log-based brittleness indices (LBI), and elastic-based brittleness indices (EBI).

The general idea of mineral-based brittleness indices suggests that mineral composition (weight fractions of different minerals contained in rock mass) governs the brittleness of the whole rock. Log-based brittleness indices imply that certain correlations exist between well logs interpretation data and brittleness. The correlations are purely empirical and are to be set based on laboratory experiments. Finally, elastic-based brittleness indices can be generally subdivided into two groups—indices can be calculated from either deformation curves from laboratory experiments on core material or data on dynamic elastic moduli. A brief overview of various brittleness indices will be provided below, but the main problem is related to the existence of a vast number of ways to introduce brittleness, some of which may contradict each other. The problem worsens as there is no way to check

whether any brittleness index is ultimately correct—while there is no universal, generally accepted brittleness index, it cannot be used to check the validity of new introduced indices.

The following general workflow is used in practice nowadays. First of all, any form of brittleness calculation is chosen to evaluate brittleness index from deformation curves obtained after carrying out laboratory tests on core samples. Hence, brittleness of rock masses is evaluated at certain points, corresponding to depths of core extractions. As the ultimate goal of brittleness estimation is the construction of brittleness index profile along the well trajectory, a certain correlation is chosen among the existing LBIs or MBIs. The parameters of the chosen correlation are obtained from the solution of optimization problem of minimizing the difference between brittleness indices obtained from laboratory tests and calculated via correlation function. The drawback of this workflow is clearly related to the ill-posed nature of the inverse problem being solved. Existence of a number of various correlations provides a possibility to obtain completely different solutions using differing correlations. These solutions can contradict each other, which leads to clear problems with using the results of brittleness estimations.

The goal of the current paper is reduction of this uncertainty: in fact, similar problems are observed while estimating elastic, strength, or transport properties of well surrounding rock masses. Rock physics approach makes it possible to incorporate information related to inner structure of rocks into the models used to reconstruct profiles of various properties from well logs and laboratory tests [13,15]. A similar approach can be developed for brittleness, and the goal of the current paper is to establish the foundation of this approach. To do that, it is necessary to understand the main factors governing brittleness via analyzing the main approaches of calculating brittleness, compare the indices proposed by various researchers, and discuss the main requirements to the workflow of brittleness reconstruction from laboratory tests on core samples and results of well log data interpretation.

The paper consists of two major parts. The following section presents a brief overview of brittleness indices generally used for hydrocarbon reservoirs analysis. Three groups of indices are analyzed, grouped with regard to type of data used for brittleness evaluation. The difference between the groups is highlighted to understand the reasons behind possible contradictions between brittleness indices calculated using different sets of data even for the same rocks. The second major part of the paper is devoted to establishing relationships between indices of different groups with the help of rock physics modeling. Original and valuable results are obtained as the relationship between rock microstructure, dynamic elastic moduli, and stress–strain response to external loading is established. In other words, rock physics modeling made it possible to find the quantitative relationship between rock microstructure and log-based brittleness index for carbonate rocks. At the same time, log-based and elastic based brittleness indices are shown to be in qualitative agreement with each other, providing insights on the methods of brittleness evaluation from geophysical data of varied scales.

## 2. Existing Brittleness Indices

There are numerous ways to introduce brittleness of rock masses. Modern review papers [3,4] highlight dozens of ways to calculate brittleness indices. Mews et al., 2019 [4] consider three major groups of brittleness indices:

1.  Mineral-based brittleness indices (MBIs) calculated from the relative fractions of different minerals contained in the rock mass;
2.  Log-based brittleness indices (LBIs) calculated from empirical correlations between particular well logs data and brittleness;
3.  Elastic-based brittleness indices (EBIs) calculated from elastic moduli characterizing the mechanical behavior of the rocks.

This classification is mainly based on the type of geophysical data used for brittleness evaluation. Not only properties of rocks in use are considered for brittleness index to be classified, but also their typical scale matters, as different types of data are related to either seismic, logging, and core scales. We will use a slightly modified version of

this classification in the current paper. According to the classification described above, certain ways of calculating brittleness of dynamic elastic moduli belong to the third group (Mews et al., 2019 [4] set them as a distinct sub-group of EBI). Nevertheless, widely applied methods of estimating brittleness from sonic logs (based on the definition suggested by Rickman et al., 2008 [16]) are purely elastic-based, yet use well logs as raw data. Therefore, we will change the third group from elastic-based brittleness indices (EBI) to test-based brittleness indices (TBI). According to this modified classification, TBIs are brittleness indices obtained from detailed analysis of deformation curves or measurements of elastic waves velocities in laboratory conditions, while LBIs use any form of well logs data interpretation, including dynamic elastic moduli profiles, as input for brittleness estimation. This modification is also needed to take scaling factor into consideration, as LBI and TBI correspond to log and core spatial scales respectively. Note that calculations of certain forms of LBIs imply usage of dynamic elastic moduli. These moduli can be obtained both from well logs and laboratory tests on core samples, so there is a direct opportunity to consider these LBIs both at well logs and core scales.

Figure 1 represents the classification of brittleness indices used in the current study. Spatial scale of brittleness index derived from each approach is added into consideration.

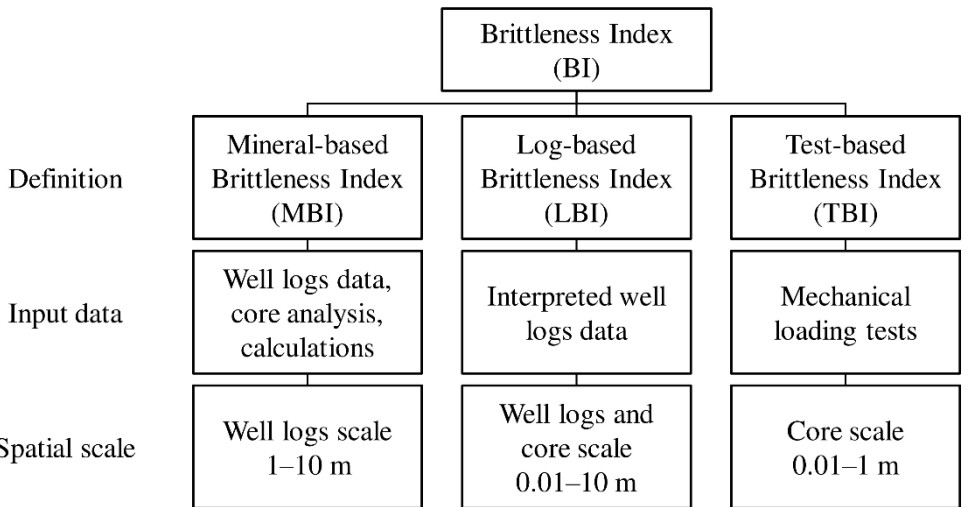

**Figure 1.** Classification of brittleness indices.

Brittleness indices have been extensively studied and discussed. Review papers by Mews et al., 2019 [4] and Zhang et al., 2016 [3] provide a lot of information on ways to deal with brittleness. We will provide brief information of different ways to calculate brittleness indices with emphasizing the inner structure and scaling factors below.

### 2.1. Mineral-Based Brittleness Indices (MBIs)

Lithological influence on brittleness of a heterogeneous rock serves as a basis to provide certain ways to compute brittleness index based on rock composition. Mineral weight faction was accepted by many researchers [17–22] as a basic parameter to evaluate brittleness. These studies reveal effects of different minerals on brittleness: e.g., presence of quartz has positive effect rock brittleness, while increase in clay fraction results into decrease of brittleness of the rock. A number of correlations have been proposed for different formations—Mews et al., 2019 [4] highlight 7 ways to calculate mineral-based brittleness indices suitable for Barnett, Neuquén Basin, Haynesville, Wolfcamp, and other shales. A general form of equation for calculating brittleness index from mineral composition was proposed by Buller et al., 2010 [23]

$$MBI = \sum_i a_i b_i M_i \Big/ \sum_i a_i' b_i' M_i. \qquad (1)$$

Here, $M_i$ stands for the weight fraction of the $i$-th mineral; $a_i$, $a_i{}'$, $b_i$, and $b_i{}'$ are empirically driven coefficients. The first pair, $a_i$ and $a_i{}'$, represent mineral specific brittleness factors: these parameters show the positive and negative effects of the particular mineral on overall brittleness respectively. The second pair, $b_i$ and $b_i{}'$, are associated with the specifics of mineral distribution in the homogeneous rock: these parameters are affected by the geometrical properties of the inner structure of the rock. Equation (1) makes it possible to take into account the effects of various minerals on brittleness of the whole composition with regard to mineral-specific effects. The majority of MBIs obtained for different fields and formations can be set using the form of Equation (1) with site-specific values of $a_i$, $a_i{}'$, $b_i$, and $b_i{}'$ factors typical for the studied rock. There are not many examples of Equation (1) being incapable of predicting brittleness of heterogeneous rocks: a notable exception has been reported by Rybacki et al., 2016 [24], who suggested introducing porosity into consideration.

There are two major problems with usage of MBIs. First of all, mineral brittleness factors $a$ and mineral distribution factors $b$ are correlation parameters. These parameters can differ from field to field a lot, so it is complicated to set a general correlation. Moreover, there is no way to measure these factors distinctively: only brittleness of the whole rock can be obtained from various kinds of tests. A certain regression problem can be postulated: consider the set of tests providing evaluations of brittleness for a number of rock samples with known mineral compositions. In fact, only weight fractions of all minerals can be obtained from independent tests, so the regression problem of establishing the proper MBI correlation for the given set of data can be deduced to finding the numbers of $a_i$, $a_i{}'$, $b_i$, and $b_i{}'$ factors in Equation (1), satisfying the mineral weight fractions and providing the MBIs corresponding to real brittleness of the studied rock samples. This regression problem is usually ill-posed: depending on the number of studied minerals and number of rock samples in the studied collection, there can be either too few or too many experimental results for obtaining the single stable solution. In the first case, there will be not enough data to obtain a single solution for the inverse problem; in the second case, the system can become overdetermined and therefore inconsistent. It is necessary to apply certain methods to obtain the proper correlation parameters for the studied set of data.

The second problem is related to the inability to use Equation (1) or its particular forms, reported in different papers, if no external data are available. Equation (1) itself does not provide a way to calculate brittleness index: external tests are necessary to obtain the factors $a_i$, $a_i{}'$, $b_i$, and $b_i{}'$, so in fact, MBIs do not provide a definition of brittleness—instead, such equations provide correlations between mineral composition and some sort of parameter that is claimed to be brittleness. Definition of brittleness is not given here—brittleness as physical property of rock can usually be defined from mechanical perspective, based on failure mechanics. Consequently, TBIs, dealing with real rock mechanical behavior under loading, provide some insight on brittleness nature.

Nevertheless, Equation (1) can provide valuable data on factors controlling brittleness. While there is no strict definition of brittleness factor and mineral distribution factor, there is still an opportunity to use them as physically meaningful parameters, rather than correlation parameters obtained from regression problem solution. In fact, the assumption of brittleness of heterogeneous rock being a function of brittleness of each mineral with certain effect from geometry of grains is naturally close to the well-developed methods of predicting other mechanical properties (starting from dynamic elastic moduli) of heterogeneous media. As a result, search of parameters in Equation (1) can be performed not from the perspective of finding the best match between calculated MBI and TBI obtained from laboratory tests. Instead, a certain mathematical model of effective brittleness of mineral composition can be constructed with respect to rock specific inner geometry. If such a model can be constructed, Equation (1) can be used to obtain MBI using data on inner structure of the rock. This will be an important step in solving the problem of BI scaling. Although it is natural to consider presence of inner inhomogeneities (e.g., natural fractures) as a factor influencing on brittleness, currently there is no way to mathematically include

such inhomogeneities into the brittleness prediction model. Nevertheless, presence of fractures will probably reveal in mineral distribution factor from Equation (1). The following algorithm can be proposed as the first step to solve brittleness index scaling problem:

1. The inner structure of rock samples is studied in detail to obtain mineral composition and geometry;
2. Rock samples are subjected to loading under laboratory condition. TBIs are obtained from the stress–strain curve (the details on this determination will be given below, in the corresponding section);
3. Data on inner structure are used to construct a mathematical model of the studied rocks. This model is used to establish the relationship between model parameters, including mineral-specific brittleness factor *a* and mineral distribution factor *b*, and brittleness obtained from loading tests. Note that mathematical model mentioned here is not obliged to be a certain rock physics model—any mathematical model, e.g., correlation dependencies—can be used for solving the problem of brittleness evaluation;
4. Upscaling procedure is established: it is expected that the established relationship between inner structure and composition of the sample and its brittleness remains valid for samples of other sizes, including bigger samples. Nevertheless, bigger samples can contain inhomogeneities of scales exceeding the typical size of the laboratory studied cores. As a result, it is necessary to include inhomogeneities of bigger sizes into the model to predict brittleness of the bigger samples;
5. An intermediate step can be completed to verify the model: if loading tests are performed on samples of different sizes, the assumption of the model being valid at any scale can be directly checked.

To sum up, mineral-based brittleness indices can be used to predict brittleness from mineral compositions of rocks using the results of laboratory tests or correlations typical for the studied formation. The problem of rescaling brittleness remains important, as data usually used for evaluating mineral weight fractions are seldom related to the same scale as results of laboratory tests.

### 2.2. Log-Based Brittleness Indices (LBIs)

Well logging data are usually used to predict a number of properties of well surrounding rock masses, including brittleness. In a situation closely resembling the discussion on MBIs, LBIs used in practice generally represent correlations for brittleness evaluation, rather than strict physical definitions of brittleness index. The question of scaling procedure remains actual as well, due to the difference between well logging data and results of experimental studies in laboratory conditions.

A number of different correlations to evaluate brittleness index profile along the well from logging data have been proposed. As it has been mentioned, porosity is an important factor influencing brittleness. As a result, while mineralogical brittleness is not correlated well with the density porosity, it is claimed to be in good correlation with the neutron porosity, leading to emergence of a set of correlations based on neutron porosity *NPHI* [20]

$$LBI_1 = \alpha NPHI + \beta, \tag{2}$$

where $\alpha$ and $\beta$ are two empirical parameters. Compressional wave slowness can be used as input data in similar linear correlations as well. Once again, empirical parameters are obtained from taking the results of laboratory tests into consideration. Some studies use compressional slowness log response *DTC* in a similar way [20]

$$LBI_2 = \gamma DTC + \delta, \tag{3}$$

where $\gamma$ and $\delta$ are still empirical parameters.

A set of methods to introduce brittleness without using laboratory tests results were proposed to overcome the need to find empirical parameters. Rickman et al., 2008 have suggested evaluating brittleness from normalized elastic moduli [16]

$$LBI_3 = \frac{E_{norm} + v_{norm}}{2}, \tag{4}$$

where $E_{norm}$ and $v_{norm}$ are normalized Young's modulus and Poisson ratio respectively

$$E_{norm} = \frac{E - E_{\min}}{E_{\max} - E_{\min}}, \quad v_{norm} = \frac{v - v_{\max}}{v_{\min} - v_{\max}}, \tag{5}$$

where $E_{\max}$ and $E_{\min}$ are the maximum and minimum Young's moduli in the formation, $v_{\max}$ and $v_{\min}$ are maximum and minimum Poisson ratios in the formation. Equation (4) was proposed for static elastic moduli, yet we still consider brittleness index $LBI_2$ a log-based, as it was obtained from initial data on dynamic elastic moduli through the correlation between static and dynamic elastic moduli. Although Equation (4) is widely used in practice for brittleness evaluation, the other LBIs have been proposed as well: Sharma and Chopra, 2012 and Sun et al., 2013 have incorporated rock bulk density $\rho$ to suggest two more brittleness indices [25,26]

$$LBI_4 = \rho E_{dyn}, \quad LBI_5 = \rho \frac{E_{dyn}}{v_{dyn}}, \quad LBI_{5'} = \frac{E_{dyn}}{v_{dyn}}, \tag{6}$$

where absolute values of dynamic elastic moduli are still obtained from well logs. $LBI_5$ and $LBI_{5'}$ are both used in practice to evaluate brittleness; they differ by taking rock density $\rho$ into account.

An attempt to incorporate mechanical failure of the rock into brittleness index determination from well logs has been made by Jin et al., 2014 [21]. Critical strain energy release rate—fracture energy independent of the applied load and geometry of the body [27]—was considered as a parameter connecting elastic moduli and failure process. It should be noted that hydraulic fracturing was the main focus of the study, so failure of planar I-mode fracture was analyzed. It was noted that critical energy release rate $G_c$ had been established for such fracture propagation, resulting in the existence of a relationship between Young's modulus and a certain fracability. The following failure mechanics equation was used to evaluate critical energy release rate [27]

$$G_C = K_{IC}^2 / E', \tag{7}$$

where $K_{IC}$ is fracture toughness, and $E'$ is equal to Young's modulus $E$ for plane stress state and $E' = E/(1 - v^2)$ for plane strain (this difference emerges from Hooke's law form for the cases of one zero principal stress—plane stress state and one zero principal strain—plane strain state). Note that dynamic elastic moduli were used in Equation (7) to predict critical energy release rate. Plane strain state was considered by Jin et al., 2014 [21] to analyze hydraulic fracture propagation. Fracture toughness was obtained from existing correlations, so well logs were used to estimate profiles of critical energy release rate. Consequently, Equation (4) was modified to incorporate fracture toughness and critical energy release rate. The following equations were proposed for brittleness evaluation [21]

$$LBI_6 = \frac{LBI_{3,norm} + G_{C,norm}}{2}, \quad LBI_7 = \frac{LBI_{3,norm} + K_{IC,norm}}{2}, \quad LBI_8 = \frac{LBI_{3,norm} + E_{dyn,norm}}{2}, \tag{8}$$

where versions of $LBI_3$, critical energy release rate, and fracture toughness are normalized as [21]

$$LBI_{3,norm} = \frac{LBI_3 - LBI_{3\min}}{LBI_{3\max} - LBI_{3\min}}, \quad G_{C,norm} = \frac{G_{C\max} - G_C}{G_{C\max} - G_{C\min}}, \quad K_{IC,norm} = \frac{K_{IC\max} - K_{IC}}{K_{IC\max} - K_{IC\min}}. \tag{9}$$

An analogue of Equation (1) can be constructed to cover the LBIs based on elastic moduli (Equations (4)–(8)): their general form can be written as

$$LBI = \sum_i c_i E_i \bigg/ \sum_i c_i' E_i,\tag{10}$$

where $c_i$ is weight factor specific for $i$-th elastic modulus $E_i$ obtained from well logs data interpretation. Dynamic moduli are used in majority of Equations (4)–(8): e.g., fracture toughness $K_{IC}$ used in Equation (7) was, in fact, obtained from a linear correlation between fracture toughness and dynamic Young's modulus typical for Barnett shale studied by Jin et al., 2014 [21]. Acquisition of weight factors remains the task of regression problem in a way similar to the one discussed above: laboratory studies on core material remain the main source of data for finding the proper correlation coefficients. Respectively, all problems discussed in the previous section remain actual, including the problem of difference between scales of laboratory tests and well logs data.

The variety of methods to calculate brittleness leaves several questions open. First of all, how does one choose the proper correlation for any formation? The majority of studies mentioned here were devoted to Barnett shale, but properties of other formations can differ a lot [16]. Secondly, it is important to know how to obtain empirical coefficients used in correlations, and, more importantly, are the results of brittleness evaluation performed by varied methods sensitive to the choice of used correlation and empirical coefficients?

Mews et al., 2019 [4] have studied these problems for four different formations: Marcellus, Chattanooga, Bakken, and Niobrara formations. Different correlations for evaluation of brittleness index were compared for these formations. Detailed study had been performed on data obtained from well logs—we would like to highlight the results obtained by Mews et al., 2019 [4]. Figure 2 represents the summary of these results: brittleness indices obtained using different correlations for the same data are shown as profiles along the wells. Certain empirical parameters and correlations were obtained with respect to the formation lithology.

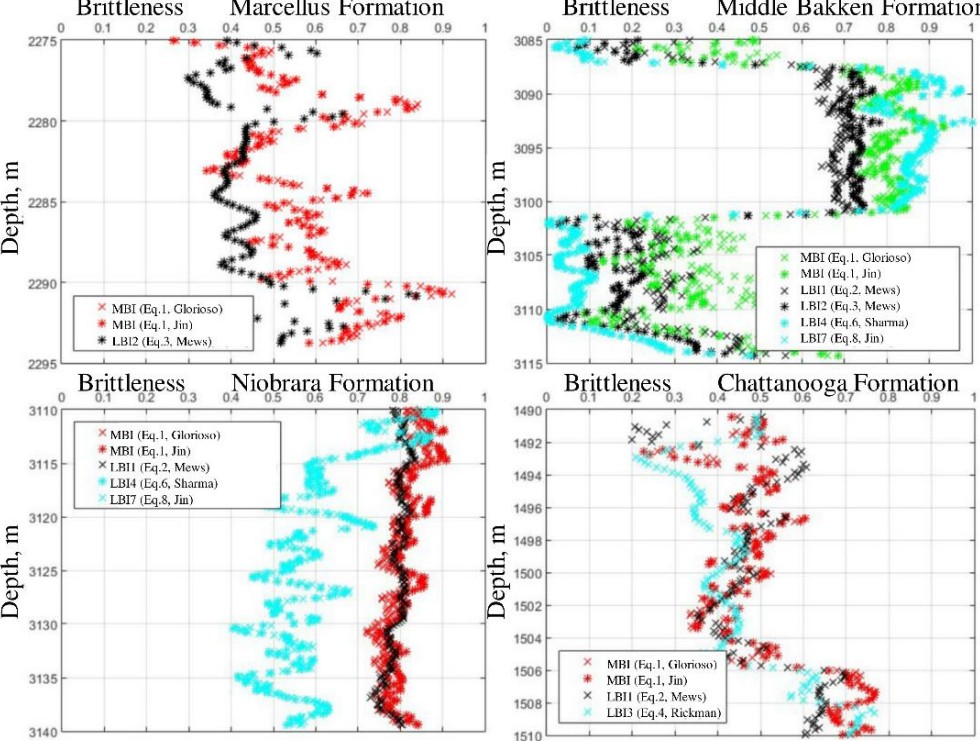

**Figure 2.** Brittleness evaluation from different correlations for four formations. Modified after Mews et al., 2019 [4].

Symbols represent predicted brittleness using the correlations proposed by authors mentioned in the legends with respect to equations in the current paper. Several important points can be seen from the figure. First of all, brittleness indices vary a lot. Normalization procedures represented by Equations (5) and (9) lead to the interval of possible brittleness indices to be between zero and one. It will be shown below that this normalization is not a tool to make things simpler, but an important way to make different indices comparable. Secondly, even within this interval, brittleness indices obtained from different correlations show varying behavior. The results obtained for middle Bakken and Niobrara formations reveal the same tendencies in different indices, i.e., zones of maximum values of different brittleness indices are in agreement. The same is not true for Marcellus and Chattanooga formations. It can be clearly seen from the figure that some brittleness indices show, in fact, opposite behavior: an example is depth interval between 2284 and 2290 m for Marcellus formation. In this interval, the zones of maximum $LBI_2$ coincide with zones of minimum MBIs, and vice versa. Moreover, this difference is not within the confidence interval: it is typical for the indices introduced in the mentioned papers to have an error of $\pm 0.1$ [21], which is lower than the differences between brittleness indices plotted in Figure 2.

Mews et al., 2019 [4] draw the following conclusion from the presented analysis: using a universal index to represent LBI or MBI results in misleading results related to selecting zones for perforation. Therefore, it is recommended that each rock type be assigned an index based on its lithology and mineralogy. Nevertheless, the question of choosing the proper brittleness index for site specific conditions remains an open question. The problem is still related to the lack of a method to check, whether the chosen brittleness index is suitable to describe failure in the particular rock. One possible answer is related to the mentioned regression problem: perhaps, the best brittleness index correlation is the one to result in the best match between brittleness evaluated from logs or mineralogy and laboratory tests. Nevertheless, there are two issues with this method of choosing a brittleness index determination process. Firstly, there still exists the scaling problem: there is no necessity for brittleness of heterogeneous rock to be the same at laboratory and well log scales. Secondly, determination of brittleness index from laboratory tests (TBI) is still a problem itself: similar to MBIs and LBIs, there are many ways to introduce TBIs from various laboratory tests on mechanical behavior of rock samples

### 2.3. Test-Based Brittleness Indices (TBIs)

Laboratory loading tests are a natural source of information regarding mechanical behavior of rocks, including their failure processes. As a result, the results of these studies can be used to evaluate brittleness of the rock samples. Once again, lack of a universal definition of rock brittleness led to existence of different ways to introduce brittleness indices calculation procedures. Zhang et al., 2016 [3] provided a comprehensive overview of brittleness indices proposed for laboratory tests analysis: we will highlight some of them below.

The majority of TBIs used in practice are related to analysis of stress–strain curve obtained from triaxial loading tests.

Figure 3 represents the typical result of single-stage loading test: a cylindrical sample is subjected to axial effective stress $\sigma_a$ and radial effective stress $\sigma_r$. Radial stress is maintained at a certain level, while axial stress is gradually increased until the failure of the sample. Differential stress $\Delta\sigma = \sigma_a - \sigma_r$ and axial and radial strains of the sample $\varepsilon_a$ and $\varepsilon_r$ are the parameters of stress–strain state of the sample obtained at any point of time during the test. Experimentally obtained axial strain dependency on differential stress plotted as a solid black line in Figure 3 is generally used to evaluate brittleness. This line has certain specific zones: linear elastic sector *OA*, inelastic strain accumulation *AB*, failure point *B*, post-peak rupture *BQ*, and relaxation zone *QK*. Specifics of behavior of the rock in these regions are used in many studies to estimate test-based brittleness indices (TBIs).

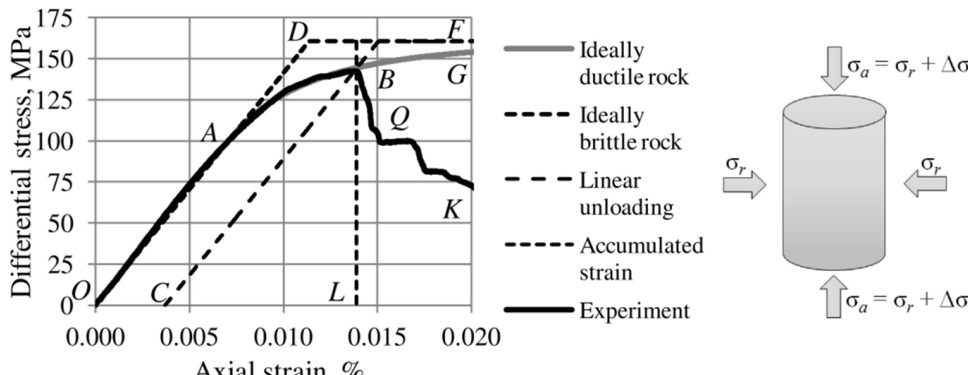

**Figure 3.** Typical single-stage loading test result for brittleness index determination. Modified after Ezhov et al., 2021 [27].

The first definitions of brittleness [6–12] have been summarized by Hucka and Das, 1974 [2] in a form related to loading test. While many definitions of brittleness were provided in a qualitative way (e.g., brittleness is lack of ductility), the following definitions were given in a quantitative form [2]

$$TBI_1 = \frac{reversible\ strain}{total\ strain} = \frac{OC}{OL}, \quad TBI_2 = \frac{reversible\ energy}{total\ energy} = \frac{S_{CBL}}{S_{OABL}} \ . \tag{11}$$

The first interpretation of the loading curve has been used here: the stress–strain curve was complemented by theoretical unloading path *BC*—potential linear unloading with a slope equal to the one observed at linear elastic loading region *OA*. As a result, the failure point can be characterized with total strain, equal to the length of segment *OC*, and its reversible part, equal to the length of segment *OL*. Simultaneously, the energetic concept can be used in a similar way: the area of curvilinear trapezoid *OABL* represents the total amount of energy stored in the sample at failure point, while the area of triangle *CBL* stands for reversible part of stored energy.

The second definition should be discussed here from energetic concept of failure process. It has already been mentioned that failure of the rock is associated with energy release: recall that critical energy release rate $G_c$ has been used for LBI calculation. In fact, within the failure mechanics concepts [28], this energy is associated with the area of free surface emerging during failure process. This free surface is, in turn, closely related to brittleness. Consider two media: very brittle and very ductile samples. A brittle sample subjected to external loading will fail with many micro fractures or one big fracture. In both cases, the cumulative area of free surfaces of all parts of the sample after failure will be large. At the same time, failure of the ductile sample will be associated with plastic flow and dislocations emergence and movement. The cumulative area of free surface of the sample after failure will be less, compared to the brittle sample. As a result, total energy stored in a ductile sample at the failure point will be higher than energy accumulated in brittle sample, as a considerable part of it was released to form free surfaces. That means that definition of brittleness index $TBI_2$ is, in fact, backed up by failure mechanics and remains in agreement with energetic concept of failure process. Various modifications of energy-based $TBI_2$ have been developed to take post-peak behavior into account [29–33]: it was shown that post-peak rupture of the sample has an important effect on rock brittleness. There are still other methods to evaluate brittleness index from results of special tests [34–39], their comprehensive review was provided by Zhang et al., 2016 [14], but their consideration is not within the scope of the current study.

There are several established ways to calculate brittleness from post-peak rupture of the sample. Following the energy concept, certain operations provide the following definitions of brittleness indices [30]

$$TBI_3 = \frac{M-E}{M}, \quad TBI_4 = \frac{E}{M} \ , \tag{12}$$

where $E$ is tangential static Young's modulus obtained from the linear zone of stress–strain curve, and $M$ is unloading modulus—the slope of the sector $BQ$ in Figure 3 corresponding to post-peak rupture associated with increase in strain and decrease in differential stress.

Existence of different methods to evaluate brittleness from loading curve leads to the same problem as for log-based brittleness indices: usage of different TBIs for the same data can result into contradicting estimations. The problem of choosing the proper brittleness index determination suitable for the specific rock is still relevant. Nevertheless, contrary to the case of log-based indices, there are insights regarding the way to verify the suitability of the chosen brittleness index. A number of studies [3,33] highlight the role of confining pressure in rock brittleness. Simultaneous analysis of stress–strain loading curve and corresponding acoustic emission emerging from microfracture emergence and propagation was performed by Zhang et al., 2016 [3] to study brittle to ductile transition: it was shown that increase in confining pressure for sets of triaxial loading tests leads to a corresponding decrease in rock brittleness. Wang et al., 2020 [33] reported that this decrease is typical for rocks of different mineral compositions. As a result, there is an opportunity to check whether a suggested TBI is valid to evaluate brittleness in a specific rock by checking its dependence on confining pressure.

The other actual problem is related to normalization of brittleness. Equation (11) are physically consistent, but their direct usage to compare brittleness indices of completely different media can be complicated, as there is no strict definition of what do minimum and maximum values of TBI mean. Ezhov et al., 2021 [27] proposed an additional normalization of brittleness: any rock sample was considered with respect to its ideally ductile and ideally brittle analogs (lines $OABG$ and $OADF$ in Figure 3 respectively). These hypothetical rock analogs were proposed in a way to represent the medium with same elastic moduli (and plastic flow potential for ideally ductile rock) as the studied sample, but with rock strength considerably higher than the studied sample. Ideally ductile analogue of the rock was considered as the potential maximum of total energy storable by the sample without failure. Plastic flow consideration made it possible to evaluate the potential strength of the sample giving an opportunity to introduce ideally brittle analog of the rock—a potential rock that deforms linearly up until its potential failure point $D$. Brittleness index of ideally ductile rock was taken as zero for TBI evaluation, and ideally brittle rock was characterized by BI of one. Hence, all brittleness evaluations were normalized with respect to these two extreme cases. Ezhov et al., 2021 [27] have shown that performance of the normalization procedure described here leads to an increase in correlation between brittleness and confining pressure obtained from loading tests performed on Bazhenov Formation rock samples. It should be mentioned that normalization procedure was established for $TBI_2$, but not to $TBI_3$ and $TBI_4$ proposed by Equation (12) to analyze post-peak behavior. The proper normalization procedure for these indices is yet to be established.

The problem of upscaling TBI was also considered for $TBI_1$: Kidybinski, 1981 [40] proposed the following procedure to evaluate effective brittleness of the layered formation: if the formation being investigated is not uniform in structure, a geological log is made, and for each layer of significance, a sample is taken and brittleness is determined in the laboratory. An average brittleness index value for the whole vertical section is then calculated as follows [40]

$$TBI_{aver} = \sum_i h_i TBI^i \Big/ \sum_i h_i, \tag{13}$$

where $E$ is tangential static Young's modulus obtained from the linear zone of stress–strain curve, and $M$ is unloading modulus—the slope of the sector $BQ$ in Figure 3 corresponding to post-peak rupture associated with increase in strain and decrease in differential stress. Is it worth mentioning that this result is similar to well-known Backus averaging used for layered formations [41].

The final remark is devoted to anisotropy of rocks: it is clear that results of laboratory tests performed at oriented samples can be used to evaluate brittleness indices in different directions. Nevertheless, there are limited ways to incorporate these oriented measurements

into discussed LBIs and MBIs: Luan et al., 2014 [42] have proposed a method to evaluate brittleness with respect elastic moduli anisotropy. LBI$_5$ was used to obtain brittleness from elastic moduli measured in the laboratory—it is a complicated task to properly incorporate energy concept into brittleness determination.

To sum up, laboratory tests provide an opportunity to evaluate brittleness of rock samples with regard to physically meaningful concepts. Yet there are still various ways to introduce brittleness index from experiments performed on rock samples, there is an opportunity to check whether the suggested index is suitable for the studied rock. The criterion is related to presence of correlation between brittleness and confining pressure. There are problems of proper brittleness normalization and upscaling: they are solved for the classic approach to measure brittleness from pre-peak behavior, but post-peak behavior is yet to be analyzed. Upscaling problem appears to be essential for brittleness evaluation from rocks: existing ways to reconstruct brittleness profiles using well logs or mineral composition use results of laboratory tests to establish the empirical parameters used in Equations (1)–(10). Although data used for estimation of TBIs and LBIs usually belong to different spatial scales—log scale for LBI, and core scale for TBI—there is an opportunity to solve the problem of rescaling. In fact, laboratory tests make it possible to estimate both static and dynamic elastic moduli. As a result, LBIs can be calculated using data obtained from laboratory tests, making the scales with possibility of LBI determination cover core scale as well. Yet, it is still necessary to use proper effective medium theory methods to solve the problem of upscaling. The following section provides the first ideas regarding the algorithm for brittleness upscaling, starting with analysis of the effect the inner structure of the rock (often called as microstructure) has on its brittleness.

## 3. Methodology and Materials

### 3.1. Rock-Physics Models and Methods of Effective Medium Theory

In order to analyze the effect of rock's inner structure on its brittleness a rock-physics model of the effective elastic properties should be constructed. The rock-physics model of effective physical properties is a triad consisting of the following elements: (1) model medium, (2) parameters of the model, and (3) equations relating the model parameters with the macroscopic (or effective) physical properties.

The model medium should mimic general specific features of the rock microstructure. Usually, in the model medium all inclusions are represented by ellipsoids (commonly, ellipsoids of revolution). The ellipsoid shape is characterized by the aspect ratio that is a ratio of semi-axis normal to the crack's plane to the semi-axis in the crack's plane.

The microstructure parameters include the aspect ratio of inclusions and parameters characterizing inclusions' orientation and mutual distribution in the rock volume.

The effective medium theory (EMT) provides the equations relating the macroscopic physical properties of rock with parameters characterizing the rock composition and microstructure. The problem of effective physical properties determination is related to the so-called 'many-body problem' and, in the general case, can be solved only approximately. This leads to the fact that there are many methods for calculation of the effective physical properties. Note that the effective physical properties that can be considered within the EMT approach include both elastic and transport properties. The latter are the thermal and electrical conductivity, as well as dielectric and hydraulic permeability [43,44].

When selecting a specific EMT method, the rock microstructure should be taken into account. For example, in the case of two-component rock of 'matrix-inclusion' type with completely isolated inclusions the upper Hashin-Shtrikman (HS) bound [1,45] of effective medium theory provides the best results if the matrix is the stiffest component. In the opposite case (stiff components in a soft matrix) the lower HS bound is preferable. Commonly, in sedimentary rocks the voids (pores and cracks) are connected. In this case, the rock-physics methods that take into account the connectivity should be applied. Among these methods are the self-consistent (SC) method [43,44,46], differential effective medium or DEM [47], *f*-model of generalized singular approximation method [13]. The SC and

DEM methods take into account the void's connectivity implicitly whereas the *f*-model incorporates a special parameter characterizing a degree of pore/crack connectivity. Strictly speaking, before applying a method of effective medium theory it should be tested on a composite material containing heterogeneities of known physical properties, shape, and distribution in the composite volume. The testing for HS bounds and *f*-model is presented in [48]. However, it is very difficult to produce such a test for SC and DEM since no explicit geometrical structure can be put in correspondence to some mental procedures accompanied the derivation of formulas of these methods. Thus, in the SC method, it is assumed that each heterogeneity can be represented by ellipsoid. Each ellipsoid is placed in a matrix having already effective properties. In other words, each inclusion is placed in a conglomerate made of other inclusions. In the DEM the inclusions are added in the matrix by portions controlled by a differential equation. Meanwhile, numerical tests show that composites having the same number of cracks with the same contact area, but differing in the number and length of the contacts, exhibit different elastic properties [49]. Besides, in real rocks, the shape of mineral grains, pores, cracks, and other inclusions is not strictly ellipsoidal. However, this is the only shape that can obtain an analytical solution for effective physical properties.

The voids in rocks often exhibit different size and relative opening (characterized by aspect ratio). In addition, in the case of polymineral rocks, each mineral component may contain a specific type of voids. These facts lead to the construction of rather complex rock-physics models containing more than one type of voids, which increases the number of unknown parameters [50–53]. This modeling makes sense if a detailed microstructure analysis is performed thereby imposing constrains to the sought-for microstructural parameters. If this is not the case, then the model parameters serve only as fitting parameters providing an acceptable correspondence between the theoretical and experimental data. In addition, the number of unknown parameters should not be sufficiently greater than the number of experimental data. Otherwise, the domain of possible solutions will be quite wide. Usually, these models are used to analyze how different pore/crack types affect the effective physical properties or to simultaneously model various effective physical properties (elastic wave velocities, electrical and thermal conductivity, etc.).

Obviously, the simplest rock-physics model of an isotropic sedimentary rock is a polycrystalline mineral matrix and randomly oriented inclusions having the same aspect ratio and filled with the same fluid. In this case, the aspect ratio serves as a parameter describing 'average' void's opening. If a rock contains both pores and cracks this parameter shows lower values compared to the case when cracks are absent.

### 3.2. Rock-Physics Model of Effective Elastic Properties Used for LBI Analysis

We apply the simplest rock-physics model described above to analyze how the porosity and pore/crack opening affect the LBI of carbonate rocks. We consider a polycrystalline calcite isotropic matrix and randomly oriented ellipsoidal voids saturated with formation water. All voids have the same shape and are filled with the same fluid. The void's aspect ratio is supposed to vary in a very wide range: from 0.0001 to 1. The void's volume concentration changes from 0 to a limiting value. For spherical pores (aspect ratio is 1) the limiting porosity is 40%. The limiting porosity value for other voids is chosen such that the crack density is equal to unity. The crack density in the effective medium theory is specified by the formula $\varepsilon = \frac{3\phi}{4\pi\alpha}$, where $\alpha$ is the aspect ratio and $\phi$ is the void's volume concentration. The crystalline matrix is an isotropic polycrystal of calcite characterizing by the elastic wave velocities, compressional ($V_p$) and shear ($V_s$), equal to 6.54 and 3.35 km/s and density 2.71 g/cm$^3$ [54]. The properties of formation water are $V_p$ is 1.6 km/s and density is 1.1 g/cm$^3$.

To relate the model parameters with the effective elastic properties of rock, we use the classical self-consistent method of the effective medium theory [43,44]. In the effective medium theory, the effective stiffness tensor $\mathbf{C}^*$ relates the stress and strain fields averaged over the representative volume of the heterogeneous medium via the Hook's law

$(\langle \sigma_{ij} \rangle = C^*_{ijkl} \langle \varepsilon_{kl} \rangle$, $i$, $j$, $k$, $l$ = 1, 2, 3). According to the self-consistent method, the formula for the effective stiffness tensor the has the form (in the tensorial form) [43,44]

$$\mathbf{C}^* = \left\langle \mathbf{C}(\mathbf{x})[\mathbf{I} - \mathbf{g}(\mathbf{x})(\mathbf{C}(\mathbf{x}) - \mathbf{C}^*)]^{-1} \right\rangle \left\langle [\mathbf{I} - \mathbf{g}(\mathbf{x})(\mathbf{C}(\mathbf{x}) - \mathbf{C}^*)]^{-1} \right\rangle^{-1}, \tag{14}$$

where the angular brackets indicate the volume averaging; $\mathbf{x}$ is a point within a representative volume whose physical properties coincide with the properties of the rock; $\mathbf{C}(\mathbf{x})$ is the stiffness tensor of a heterogeneity at the point $\mathbf{x}$; $\mathbf{I}$ is the unit tensor of the fourth rank having the components $I_{ijkl} = \frac{1}{2}\left( \delta_{ik}\delta_{jl} + \delta_{il}\delta_{jk} \right)$, $i$, $j$, $k$, $l$ = 1, 2, 3. The volume averaging in Equation (14) is performed over all components with the use of their volume concentrations. In this method the ellipsoidal shape is assumed for all heterogeneities. The tensor $\mathbf{g}$ depends on the shape of ellipsoidal heterogeneities and the effective properties. This tensor has the form [44,55]

$$
\begin{aligned}
&g_{kmln} = -\tfrac{1}{4}\left( \widetilde{a}_{klnm} + \widetilde{a}_{mlnk} + \widetilde{a}_{knlm} + \widetilde{a}_{mnlk} \right), \\
&\widetilde{a}_{kmln} \equiv \tfrac{1}{4\pi} \int_0^{2\pi} \int_0^{\pi} n_{mn} \Lambda_{kl}^{-1} d\Omega, \; d\Omega \equiv \sin\theta d\theta d\varphi, \\
&\Lambda_{kl} \equiv C^*_{kmln} n_{mn}, \; n_{mn} \equiv n_m n_n, \\
&n_1 = \tfrac{1}{a_1} \sin\theta \cos\varphi, \; n_2 = \tfrac{1}{a_2} \sin\theta \sin\varphi, \; n_3 = \tfrac{1}{a_3} \cos\theta,
\end{aligned}
\tag{15}
$$

where $a_1$, $a_2$, and $a_3$ are semi-axes of ellipsoids describing the shape of heterogeneities. Commonly, the shape of heterogeneities is modeled by ellipsoids of revolution with $a_1 = a_2 = a$ and $a \geq a_3$ (oblate spheroids). In this case the shape can be characterized by an aspect ratio that is equal to $a_3/a$. As seen Equation (14) contains the term $\mathbf{C}^*$ in the both left- and right-hand sides. In this case, the solution is found by iterations.

For isotropic rocks, the compressional ($V_p$) and shear ($V_s$) wave velocities are calculated from the effective stiffness tensor $C^*$ (in Voigt matrix representation) and density of rock $\langle \rho \rangle$ as

$$V_P = \sqrt{\frac{C^*_{11}}{\langle \rho \rangle}}, \; V_S = \sqrt{\frac{C^*_{44}}{\langle \rho \rangle}}. \tag{16}$$

The rock density is equal to the volume average of the densities of all components, and this averaging gives an exact solution for the density.

Equation (14) is more complex than the self-consistent formulas of Berryman [46], which are applicable only if the properties of all components and the effective properties are isotropic. However, Equation (14) is sufficiently unified since in general it is applicable for anisotropic effective properties of any type of symmetry. The components may have anisotropic elastic properties, which suggests that inclusions can be solid as well. This choice of the method for the LBI analysis gives us an opportunity to consider unconventional anisotropic reservoir rocks in our future analysis, including shales or carbonate rocks with oriented fractures. Besides, unified Equation (14) allows us to analyze how oriented fractures arising in an isotropic rock change the rock brittleness. Note that Equation (14) and self-consistent formulas of Berryman give the same results for spherical inclusions.

When applying Equation (14) for analysis of brittleness the shape of mineral matrix particles is assumed to be spherical. The averaging is performed for two components—spherical pieces of mineral matrix and voids saturated with formation water. We consider randomly oriented voids which leads to isotropic effective properties. For the calculations, we use our own Fortran code that has previously been thoroughly tested.

## 4. Results

### 4.1. Testing of Rock-Physics Model Selected for LBI Analysis on Experimental Data

Before applying the constructed rock-physics model for the LBI analysis of carbonate rocks we test it on available log data on elastic wave velocities, porosity, and density. For each log depth, we construct a function having the form

$$\psi = \left[ a \frac{\left| V_P^{calc} - V_P^{\exp} \right|}{0.5\left( V_P^{calc} + V_P^{\exp} \right)} + b \frac{\left| V_S^{calc} - V_S^{\exp} \right|}{0.5\left( V_S^{calc} + V_S^{\exp} \right)} \right] 100\% \qquad (17)$$

This function is a linear combination of relative differences between theoretical (*calc*) and experimental (exp) velocities $V_P$ and $V_S$. In view of high quality of the shear wave velocity in the used log data we put $a = b = 0.5$ in the function $\psi$. The rock-physics model is considered applicable if we can find an aspect ratio that gives a value of function $\psi$ not exceeding 10% (we call it acceptable value). This value usually characterizes the log data quality. Note that many solutions for aspect ratio can be found that provide an acceptable value of $\psi$. Among the found acceptable solutions we choose only the one that satisfies the minimum of function $\psi$ (the most probable solution) for the current log depth.

Figure 4 shows aspect ratios inverted from log data (2371 depth points) of a well penetrating carbonate reservoir rocks (limestones) of West Siberia (Russia). Another set of aspect ratios shown in Figure 4 is derived from well logs of limestone layers within a shale formation [54]. We select only the depths where calcite volume concentration is at least 90% (17 depth points). Figure 5 demonstrates the quality of the solutions found. The horizontal axis shows the experimental velocities and the vertical axis demonstrates the relative difference between the experimental and theoretical velocities. The negative values mean that the theoretical velocity is lower than the experimental. As seen, for the both data sets, the misfit between the theoretical and experimental velocities is within 10% excepting two points for the second dataset.

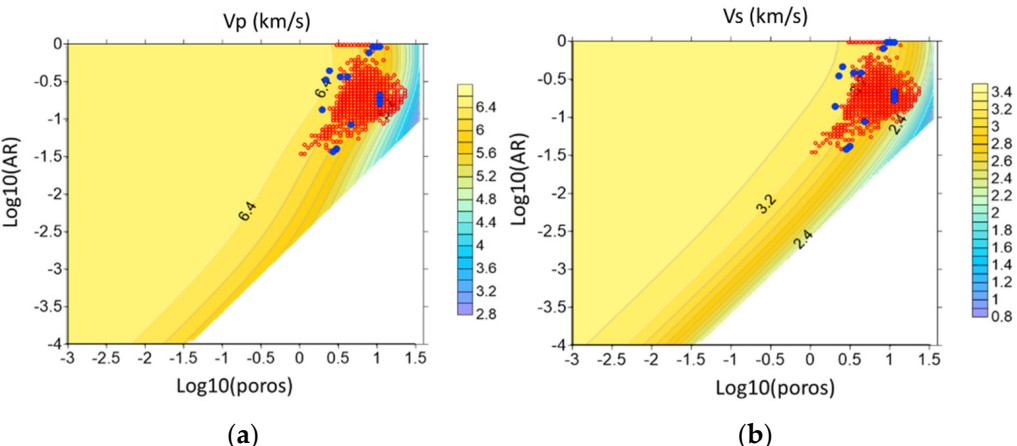

(**a**)         (**b**)

**Figure 4.** (**a**) Compressional and (**b**) shear wave velocities for a model of calcite matrix with randomly oriented brine-saturated voids versus decimal logarithms of porosity (poros) and void's aspect ratio (AR). Red signs show values for West Siberian limestones. Blue signs are used for limestones from layers within a shale formation [54].

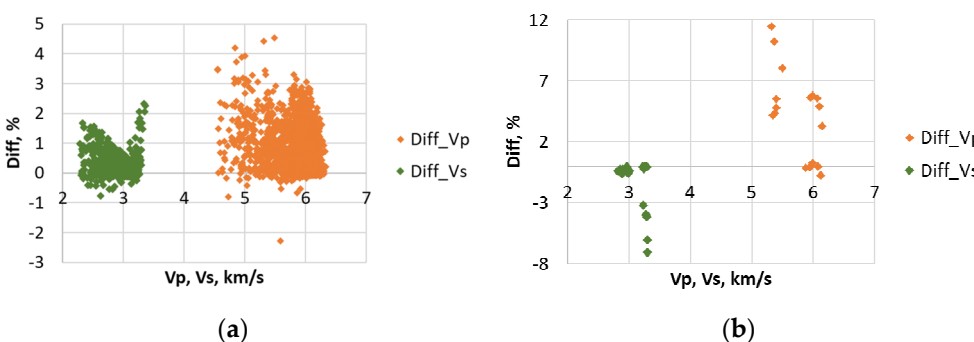

**Figure 5.** Relative difference (Diff) between the theoretical and experimental values of compressional (orange signs) and shear (green signs) wave velocities. (**a**) West Siberian limestones and (**b**) limestones from layers within a shale formation [54]. The horizontal axis shows experimental velocity values.

As seen even a simplest rock-physics model can be used to explain a behavior of elastic wave velocities of carbonate rocks (limestones) at the log scale. However, the modeling with constant elastic moduli of mineral matrix may be not too successful for laboratory data. According to our experience, this model can produce overestimated compressional wave velocities. This is explained by the fact that carbonate rocks contain closed porosity. As a rule, the log data give the total porosity (from density logs) whereas in laboratory the porosity is measured by the sample saturation. As a result, the laboratory measurements of porosity characterize effective (or connected) porosity instead of the total porosity. This means that rock-physics modeling at the core scale should include the "matrix" elastic moduli in the list of unknown parameters considering the "matrix" as a part of the rock that does not participate in the fluid motion.

### 4.2. Effect of Microstructure Parameters on Brittleness Indices

Figure 6 exemplifies the Young's modulus and Poisson ratio for a model with brine-saturated randomly oriented voids versus the characteristics of pore space—decimal logarithms of void's porosity (porosity is in %) and aspect ratio. The density varies from 2.71 to 2.17 g/cm$^3$ in the porosity range from 0 to the limiting value that depend on the aspect ratio (as described in the previous section). As expected, the Young's modulus decreases as the porosity increases for a fixed aspect ratio. If porosity is a constant value, the Young's modulus increases with the increase in aspect ratio. For the Poisson ratio, the opposite behavior is observed. A rather wide domain for almost constant Poisson ratio (around 0.32–0.33) is seen.

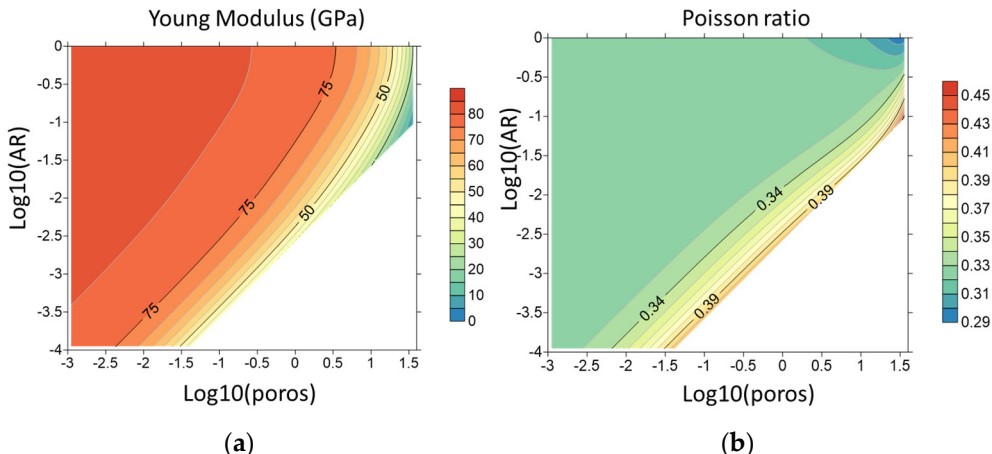

**Figure 6.** (**a**) Young's modulus and (**b**) Poisson ratio for a model of carbonate matrix with randomly oriented brine-saturated voids versus decimal logarithms of porosity and void's aspect ratio.

Figure 7 demonstrates the dependences of indices on the pore space characteristics.

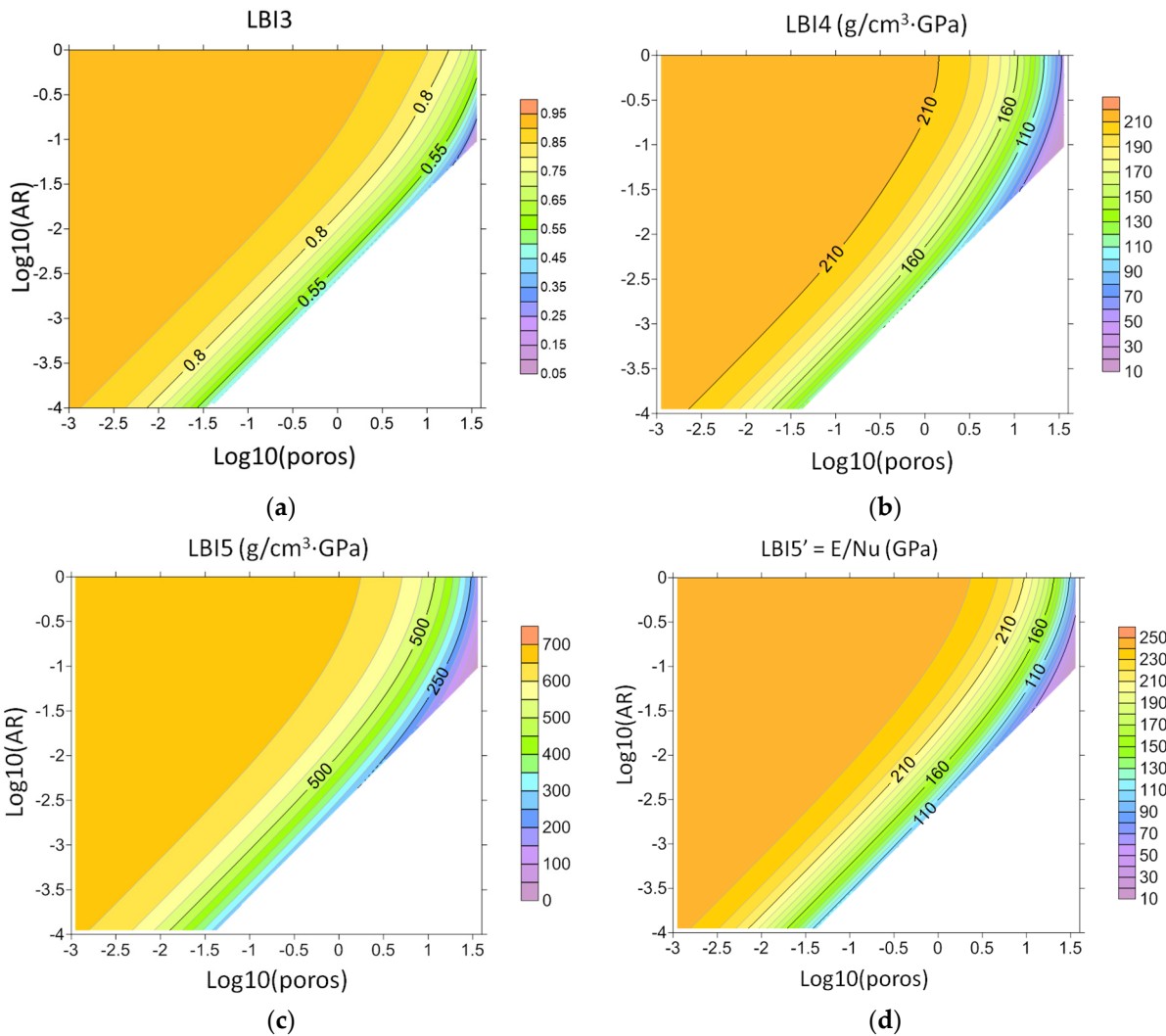

**Figure 7.** Brittleness indices (**a**) $LBI_3$, (**b**) $LBI_4$, (**c**) $LBI_5$, and (**d**) $LBI_{5'}$ for a model of carbonate matrix with randomly oriented brine-saturated voids versus decimal logarithms of porosity and void's aspect ratio.

As seen, all the indices show a similar behavior. They decrease as the porosity increases and increases with void's opening, which is in agreement with $LBI_1$ behavior. Among the indices, the $LBI_5$ is more variable whereas the smallest variability is observed for $LBI_3$. Similar behavior is observed for $LBI_8$ (Figure 8). Its variability is comparable to $LBI_3$.

In order to calculate the indices $LBI_6$ and $LBI_7$, first, the fracture toughness and critical strain energy release rate should be estimated. To obtain the fracture toughness we apply the regression equation from the work [56] that has the form $K_{IC} = 0.3 + 0.027E$, where the fracture toughness $K_{IC}$ is given in MPa $\times \cdot m^{0.5}$, and $E$ is the Young's modulus (in GPa). The critical strain energy release rate is calculated by Equation (7). The fracture toughness and the critical strain energy release rate versus the pore space characteristics are shown in Figure 8. Since the relation between the fracture toughness and Young's modulus is linear, the behavior of fracture toughness is similar. The strain energy release also decreases with the porosity and increases with the void opening.

Figure 9 demonstrates similar dependences for $LBI_6$, $LBI_7$, and $LBI_8$. As seen, for indices $LBI_6$, $LBI_7$ the behavior is different as compared to other ones. The indices increase with the porosity. For small opening of voids (not greater than 0.01) the indices decrease with the opening. For more open voids the behavior of indices becomes nonmonotonic. It could be explained by that the regression equation used for the fracture toughness

estimation is applicable for rather thin voids. The index $LBI_6$ is more variable compared to $LBI_7$.

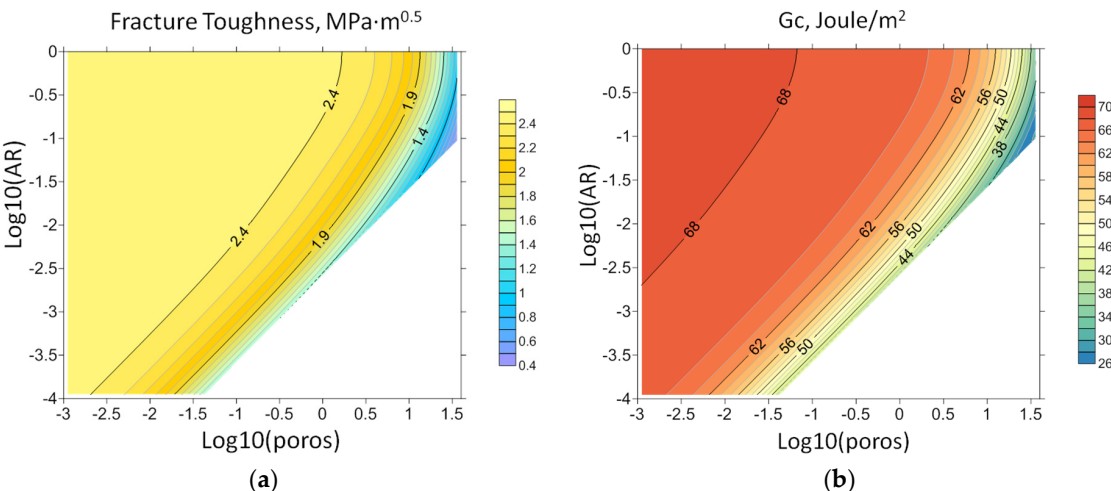

(a)

(b)

**Figure 8.** (**a**) Fracture toughness and (**b**) critical strain energy release $G_C$ versus decimal logarithms of porosity and void's aspect ratio.

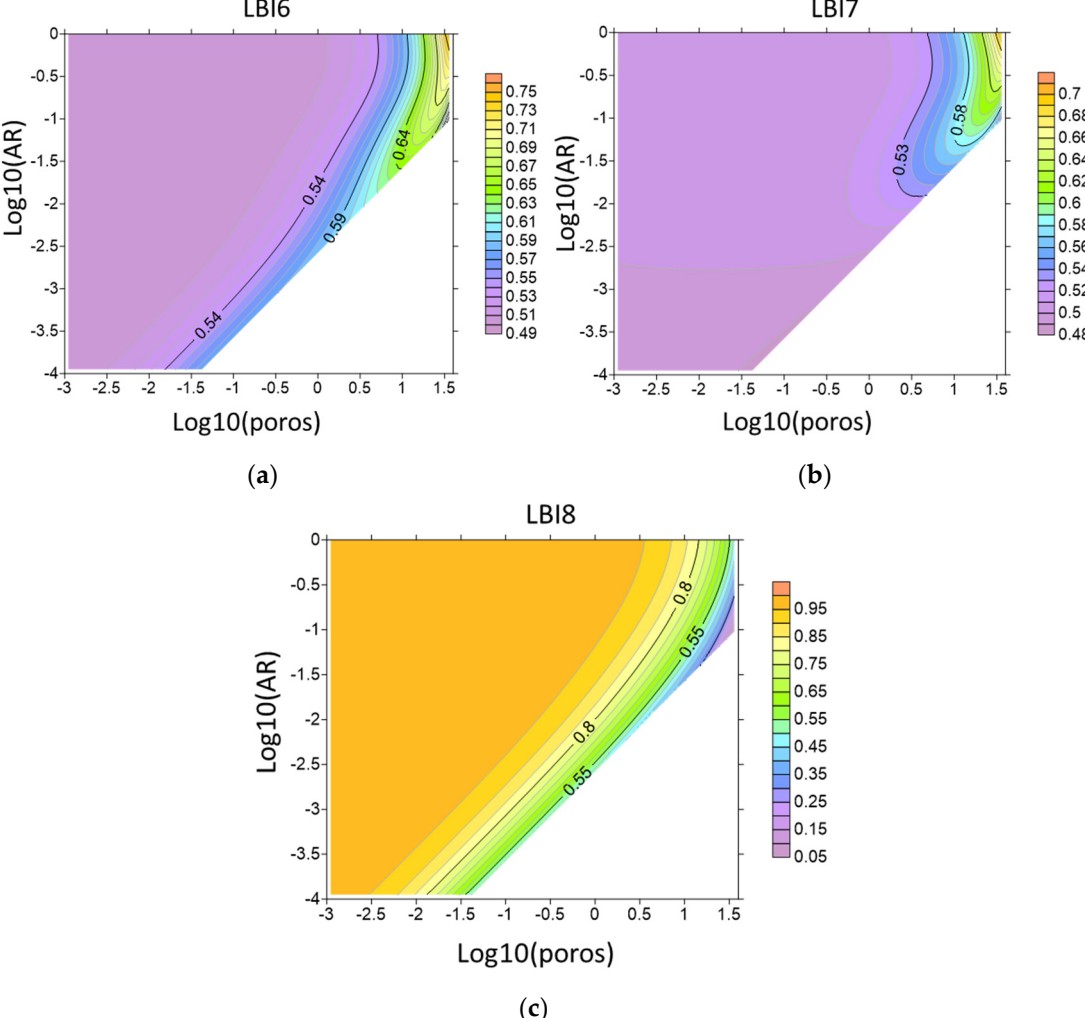

**Figure 9.** Brittleness indices (**a**) $LBI_6$, (**b**) $LBI_7$, and (**c**) $LBI_8$ for a model of carbonate matrix with randomly oriented brine-saturated voids versus decimal logarithms of porosity and void's aspect ratio.

## 5. Discussion

Behavior of LBI$_6$ deserves particular discussion. Recall that LBI$_6$ was proposed with energy concept kept in mind: critical energy release rate was considered as a factor controlling brittleness. Energy concept has been used for introduction of TBIs as well, so it can be used to establish a bridge between TBIs and LBIs. This link can be highlighted after considering of inner structure influence on failure process. Figure 10 represents brittleness index LBI$_6$ as function of fractures density. Note that the fracture density $\varepsilon$ is related to the number of fractures $N_{fr}$ by the linear equation $N_{fr} = V\varepsilon/a^3$, where $V$ is the volume containing the fractures, and $a$ is the fracture's half-length.

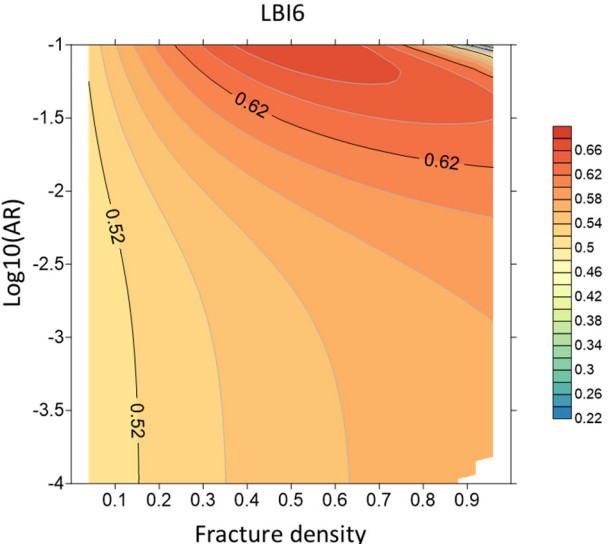

**Figure 10.** Brittleness index LBI$_6$ versus fracture density and decimal logarithm of aspect ratio.

It can be seen that increase in the fracture density and, as a result, in the number of natural fractures in the rock volume leads into an increase in its brittleness.

This effect can be explained by detailed analysis of failure process. Each fracture serves as a stress concentrator during loading. Increase in stresses leads to development of natural fractures—the process associated with increase in total free surface existing in fractured rock. Hence, the more fractures are present, the more free surface emerges during loading. As a result, more energy is released during brittle failure. Hence, increase in relative amount of fractures—or fracture density in terms of rock physics modeling—is expected to lead to increase in brittleness index TBI$_1$ or TBI$_2$. At the same time, rock physics modeling has been carried out for calculating LBIs, so the tendency shown in Figure 10 is in fact visualization of the connection between energetic concept of brittleness, rock physics model, and log-based empirical correlation for brittleness index evaluation.

This rock physics approach can be also applied for changing mineral composition and pore-filling substance thereby providing a link between MBI and LBI indices.

Note that this approach for LBI analysis in terms of rock composition and microstructure can be applied at any scale of interest—core, log, or seismic scale. For example, if we have other types of heterogeneities at the log scale, which are much greater than those at the core scale, we should introduce them in the rock physics model considering the effective properties calculated at the core scale as a background medium with effective properties. Thus, subvertical fractures tens of centimeters in size can exist in carbonate rocks or elongated lenses of kerogen or clay. In this case, the rock physics modeling is again suitable for an analysis shown above but for anisotropic rocks of HTI (horizontal transverse isotropy) or VTI (vertical transverse isotropy) symmetry. Note that Equation (14) of self-consistent method is general and applicable to anisotropy of any type not only for HTI or VTI. We try to perform such an analysis and discover that a lot of questions arise when calculating the LBIs to anisotropic rocks. The only exception is LBI$_4$ that is rather

simple, linear dependent on the Young's modulus, and does not require normalization. In this case, we can compare the indices calculated for different directions and distinctly conclude that the index in the symmetry plane is greater than that in the perpendicular plane. However, when considering $LBI_{5'}$ or $LBI_3$ that are widely used in practice or other indices the comparison meets some difficulties since it is not clear which minimum and maximum of moduli should be used for the normalization—over all directions or only for the particular direction. The same problem arises when dealing with TBIs—although it is natural to treat TBIs obtained from samples with known spatial orientations as anisotropic indices, construction of a single parameter standing for brittleness in all directions remains a challenge. There are attempts to construct a tensor characterizing brittleness anisotropy [42], but a strict definition of anisotropic brittleness of tensor nature is yet to be established. These problems should be analyzed in more detail and this is out of scope of this paper.

Several remarks should be made for practical use of the results. If a rock is highly brittle it is good for fracturing and bad for drilling, and vice versa. This means that brittleness concept has opposite meaning for these two operations, as highly brittle rocks are preferable in some conditions, and not preferable in others. Nevertheless, brittleness estimation remains essential for many problems of hydrocarbon reservoir exploration and development, regardless of likeability of brittle rock for the particular situation. Our results obtained for two limestone datasets (Section 4.2) demonstrate that almost all LBI indices (except specific $LBI_6$ and $LBI_7$ indices) have reduced values compared to the other brittleness values shown in Figures 7 and 9. This is in line with the successful drilling of the wells and high-quality log data.

## 6. Conclusions

Different empirical formulas for calculating the brittleness index exist that is due to different scale of problem consideration. Therefore, different experimental data available at the respective scale are used for the brittleness estimation. This provides a variety of formulas for brittleness definition. In this paper, the definitions are classified according to the input data and scale. Three types of BI definitions are analyzed—mineral-based (MBI), log-based (LBI), and test-based (TBI).

Generally, the brittleness depends on the mineral composition and pore space geometry. These parameters control the macroscopic elastic properties of rocks. This makes it possible to apply the effective medium theory to analyze how the composition and microstructural parameters control the brittleness indices. The rock-physics based analysis perform in this work allows us to obtain dependence of different brittleness indices of LBI type on the porosity and void's relative opening (aspect ratio). For a simple model of carbonate porous rock with randomly oriented brine-saturated voids, the indices $LBI_3$, $LBI_4$, $LBI_5$, $LBI_{5'}$, and $LBI_8$ that are calculated from the elastic moduli and density, decrease with porosity and increase with the relative opening of voids. However, the LBIs, that are dependent on the fracture toughness and critical strain energy release rate, increase with the number of fractures and their opening. The rock physics analysis of anisotropic rocks is not trivial and requires special attention.

At the same time, analysis of stress–strain relationships of the samples subjected to external loading provides a lot of information regarding brittleness as well. There are still various ways to introduce brittleness indices from laboratory tests, but there are also opportunities to check whether suggested definition is valid—e.g., via analyzing confining pressure effect on brittleness of the rock. There is also a strong physical background behind some of the definitions as TBIs are treated with respect to energy conservation law.

Rock physics modeling can be applied to establish the relationship between TBIs and LBIs defined with regard to energy concept. The results presented in the paper highlight the bridge existing between these indices ($LBI_6$ and $TBI_2$ in particular). Accumulation of plastic strains in the sample under stress can be realized through one of two ways: it is either related to slow movement of dislocations or increase in free surface of the sample, i.e., increase in number and density of the fractures. Both modeled log-based index and

discussed test-based indices increase with growth of the amount of fractures, so there is a positive correlation between these two parameters.

There are many problems with estimation of brittleness for practical needs: there are various ways to define brittleness itself; data used for brittleness evaluation represent different temporal and spatial scales; correlations used for brittleness estimation differ from field to field and so on. Nevertheless, proper understanding of brittleness nature and rock physics modeling provides an opportunity to check which of the ways to evaluate brittleness is the proper one for a particular field. Given the importance of brittleness prediction for such operations as hydraulic fracture design, the discussed problem deserves further analysis.

**Author Contributions:** Conceptualization, N.D. and I.B.; Methodology, N.D.; Software, I.B.; Validation, M.B.; Formal analysis, I.B.; Investigation, N.D., I.B. and M.B.; Writing—original draft preparation, N.D.; Writing—review and editing, I.B.; Visualization, N.D. and I.B.; Supervision, I.B.; Project administration, N.D. and I.B. All authors have read and agreed to the published version of the manuscript.

**Funding:** This research was performed as part of the State assignment of Schmidt Institute of Physics of the Earth RAS.

**Institutional Review Board Statement:** Not applicable.

**Informed Consent Statement:** Not applicable.

**Data Availability Statement:** Simulated data on the elastic moduli, elastic wave velocities, fracture toughness, critical strain energy release, and LBI indices are available at the link: https://cloud.mail.ru/public/KWMp/hWEiSqJGW (accessed on 22 November 2021).

**Conflicts of Interest:** The authors declare no conflict of interest.

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
