# Peer review of "Problems of Multiscale Brittleness Estimation for Hydrocarbon Reservoir Exploration and Development"

_applsci, doi:10.3390/app12031134_

Round 1
Reviewer 1 Report
The manuscript is presented in a poor way. Citations of the main references ar missed. For example, the first 3 or 4 paragraphs in the introduction section have no single Citations. However, the reality is slightly these information have been taken from external references. This is an ethical is, you cannot write the general overview of the introduction section without Citations. Whst is the basic of the classification of brittleness indices? References for all equations are missed. I novelty of the works is weekly defined. I suggest changing paper type to a review paper with re-write the manuscript, in a better way. The paper should take major revisions.
Reviewer 2 Report
This manuscript investigates the brittleness estimation using geophysical data. Three approaches are deployed in this study including mineral-based, log-based, and elastic-based brittleness estimation. The innovation part of this manuscript is the discussion on the contradictions of estimations from different approaches. The structure of this study is good, and the language is even beyond my level.
I enjoy reading this paper, but the validation part of this study is relatively poor.
I advise that the validation part should be added in the discussion part. The detailed comment is as following:
The concept of brittleness is even different for the drilling and fracturing field in the Petroleum Production. For instance, the brittleness of coal is supposed very good for drilling engineering. However, the coal is very hard to be fractured (show obviously plasticity) for the fracturing engineering. How do your study solve this contradiction? Hence the application of this study should be provided for a specific reservoir, such as shale, coal or tight sand. This could show the applicability and practicality of this study especially in this journal.
Reviewer 3 Report
Dear author:
I have read the article entitled "Problems of multiscale brittleness estimation for hydrocarbon reservoir exploration and development" written by Nikita Dubinya, Irina Bayuk, and Milana Bachmach.
The main objective of the study is focused on the problem of using geophysical data to estimate brittleness of rock masses for the needs of petroleum industry.
The article should be organized according to the standards of current journals: Introduction, 2 Methodology and Materials, 3 Results, 4 Discussion and 5 Conclusions.
The figures must include the reference in the event that they are not made by the author.
Line 52 should include references.
Chapter 1. Introduction
The introduction is a review text of the state of the art and an exposition of the purpose of the paper.
This chapter should be redrafted, as there are repeated concepts in various parts of the text. Some expressions should not be present in the text (line 159, line 332).
Chapter 2. This section is well developed, but should be better structured as there is a mix of reference check information.
Part of the text introduces the research carried out by the authors. In my way of understanding they should do a section on Methodology and materials.
They must explain in more detail the model they apply and the software used. They must include the number of samples tested and from which the data shown in the graphs represented is obtained.
The authors have done a good job of reviewing the state of the art based on the literature referenced in References.
The references are well developed according to the publisher's regulations.
You need a great review of the exposed information. They should give a structure to the text including the information provided by the authors with more details.
Best regards
Round 2
Reviewer 1 Report
The paper is accepted for publication in its current form.